# An unexpected strategy to alleviate hypoxia limitation of photodynamic therapy by biotinylation of photosensitizers

Jing An[1], Shanliang Tang[1], Gaobo Hong[1], Wenlong Chen[1], Miaomiao Chen[1], Jitao Song [2], Zhiliang Li[2], Xiaojun Peng[1], Fengling Song [1,2✉] & Wen-Heng Zheng[3✉]

The most common working mechanism of photodynamic therapy is based on high-toxicity singlet oxygen, which is called Type II photodynamic therapy. But it is highly dependent on oxygen consumption. Recently, Type I photodynamic therapy has been found to have better hypoxia tolerance to ease this restriction. However, few strategies are available on the design of Type I photosensitizers. We herein report an unexpected strategy to alleviate the limitation of traditional photodynamic therapy by biotinylation of three photosensitizers (two fluorescein-based photosensitizers and the commercially available Protoporphyrin). The three biotiylated photosensitizers named as compound **1**, **2** and **3**, exhibit impressive ability in generating both superoxide anion radicals and singlet oxygen. Moreover, compound **1** can be activated upon low-power white light irradiation with stronger ability of anion radicals generation than the other two. The excellent combinational Type I / Type II photodynamic therapy performance has been demonstrated with the photosensitizers **1**. This work presents a universal protocol to provide tumor-targeting ability and enhance or trigger the generation of anion radicals by biotinylation of Type II photosensitizers against tumor hypoxia.

[1] State Key Laboratory of Fine Chemicals, Dalian University of Technology, 116024 Dalian, China. [2] Institute of Molecular Sciences and Engineering, Institute of Frontier and Interdisciplinary Science, Shandong University, 266237 Qingdao, China. [3] Department of Interventional Therapy, Cancer Hospital of Dalian University of Technology, Liaoning Cancer Hospital and Institute, 110042 Shenyang, China. ✉email: songfl@dlut.edu.cn; Mir2yue2@163.com

Photodynamic therapy (PDT) is an emerging alternative treatment modality for malignant tumor[1–3], which bases on photosensitizers to transfer light energy into reactive oxygen species (ROS) to induce cell apoptosis and tissue damage[4–6]. Unfortunately, the therapeutic efficacy of PDT is limited by the hypoxic environment in solid tumors[7,8]. Because the dominant mechanism of PDT is based on high-toxicity singlet oxygen ($^1O_2$), which is called type II PDT[9,10]. Type II PDT highly relies on the surrounding oxygen[11,12], which is in conflict with the inherent properties of tumor hypoxia. Fortunately, recently Type I PDT has been found to be able to perform well under a hypoxic environment[13–15]. In contrast to the direct energy transfer from excited photosensitizers to $O_2$ in the Type II pathway, the Type I process is that the excited photosensitizers transfer electrons or hydrogen protons to the surrounding substrates, thereby yielding radical species (e.g., superoxide ($O_2^{-\bullet}$) and hydroxyl (OH•) radicals)[16,17]. Among these radicals, excessive $O_2^{-\bullet}$ is known to be able to react with proteins, DNA, and lipids, causing irreversible damage to cellular components[18,19]. Furthermore, $O_2^{-\bullet}$ could then participate in superoxide dismutase (SOD) - mediated disproportionation reactions, which would realize the reuse of $O_2$ and trigger the formation of other highly toxic ROS[20].

The light source is another important component in photodynamic therapy. White-light PDT has been proposed as an effective treatment for fungal diseases and dermatological lesions (like acne, keratosis, skin tumors, etc.)[21–23]. In particular, daylight photodynamic therapy (DL-PDT) has made PDT more widespread, cheaper, and less painful[24]. Although some white-light activated photosensitizers have been reported, their absorption profiles mismatch the spectral emission of the white-light source (i.e., solar radiation)[25,26]. Furthermore, the majority of those photosensitizers are only able to generate one type of ROS (e.g.,$^1O_2$)[27,28].

In this study, three organic photosensitizers with both Type I and Type II mechanisms for white-light activated PDT were developed by biotinylation of typical Type II photosensitizers. In 2018, Peng's group reported a Type I photosensitizer containing a biotin unit with $O_2^{-\bullet}$ generation[29]. And the role of the biotin unit was considered to achieve preferential tumor-targeting ability. Inspired by it, the original purpose of introducing biotin into the three photosensitizers was also to provide the photosensitizers with the ability to target tumors, as biotin receptors are over-expressed on the surface of cancer cells or tumor vasculature systems compared with normal tissues[30–35]. Unexpectedly, the $O_2^{-\bullet}$ generation capacity of all the three photosensitizers (**1**, **2**, and **3**) were found to be greatly boosted by biotinylation. Furthermore, compound **1** was studied for a combinational Type I/Type II DL-PDT, which confirmed that the biotinylation strategy can efficiently alleviate the limitation of hypoxia.

## Results

**Synthesis and photophysical properties.** Several fluorescein derivatives with thermally activated delayed fluorescence have been reported as PDT photosensitizers in our previous works[36–38]. These fluorescein derivatives were found to be Type II photosensitizers. In this work, two fluorescein photosensitizers **1** and **2** were synthesized from their precursors **4** and **5** by amide condensation with 5-(2-oxohexahydro-1H-thieno[3,4-d]imidazol-4-yl) pentanehydrazide (Biotin–NH-NH₂,**11**), respectively (Fig. 1 and Supplementary Fig. 1). In contrast, we also synthesized a porphyrin photosensitizer **3** by covalently linking compound **11**

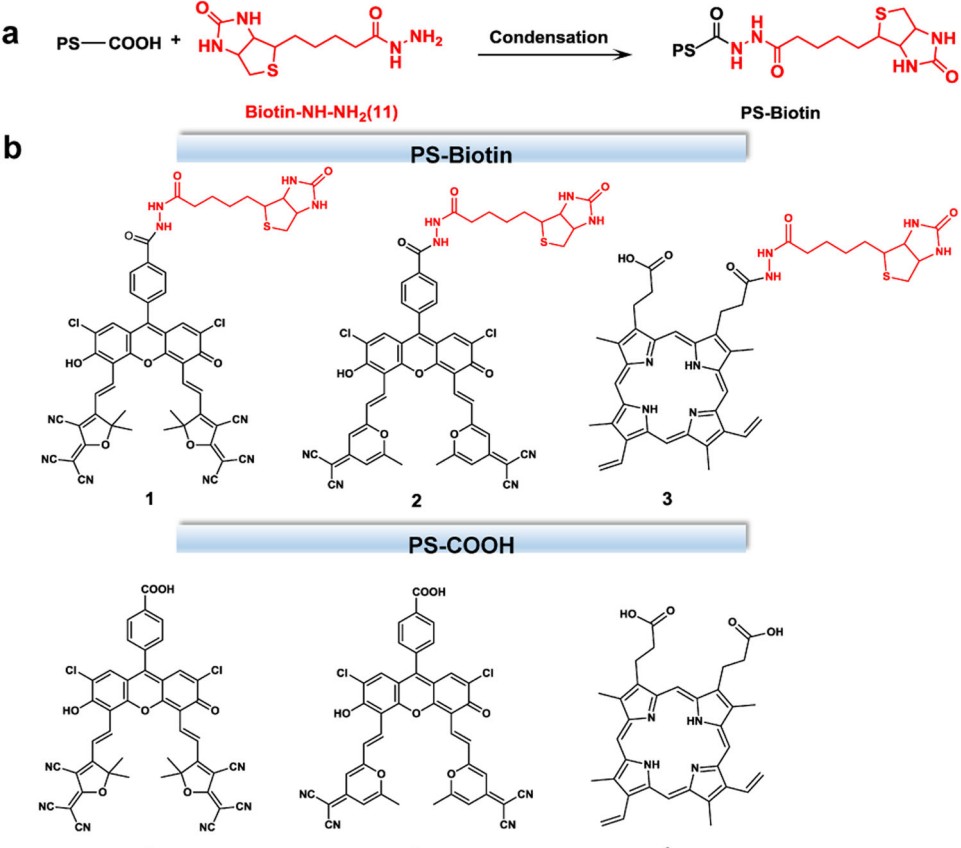

**Fig. 1 The synthesis route for biotinylation of photosensitizers from PS-COOH to PS-Biotin. a** The carboxyl-terminal photosensitizer PS-COOH is condensed with hydrazine modified biotin to form PS-Biotin. **b** Chemical structures of compounds (PS-Biotin and PS-COOH) 1–6.

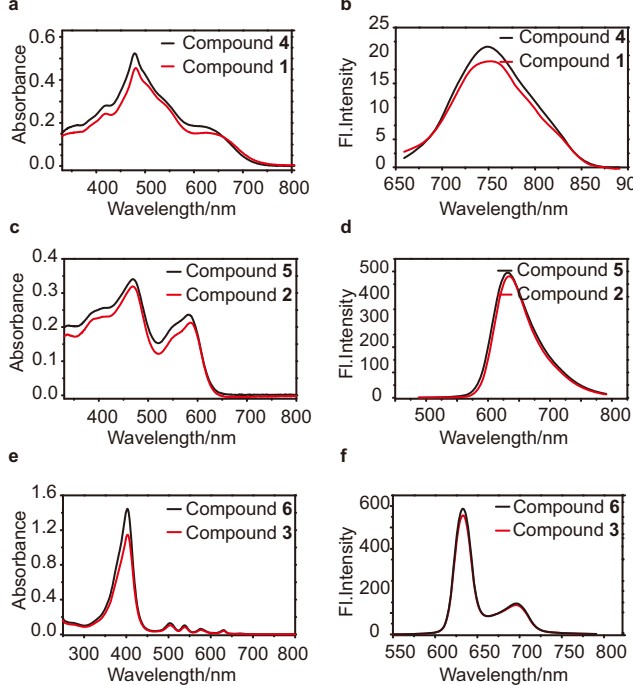

**Fig. 2 Absorption spectra and emission spectra of compounds 1–6.**
Absorption spectra of (**a**) compound **1** and **4**, (**c**) **2** and **5**, (**e**) **3** and **6**.
Emission spectra of (**b**) **1** and **4** ($\lambda_{ex}$ = 640 nm), (**d**) **2** and **5**
($\lambda_{ex}$ = 469 nm), (**f**) **3** and **6** ($\lambda_{ex}$ = 403 nm) in ethanol. Fl fluorescence.

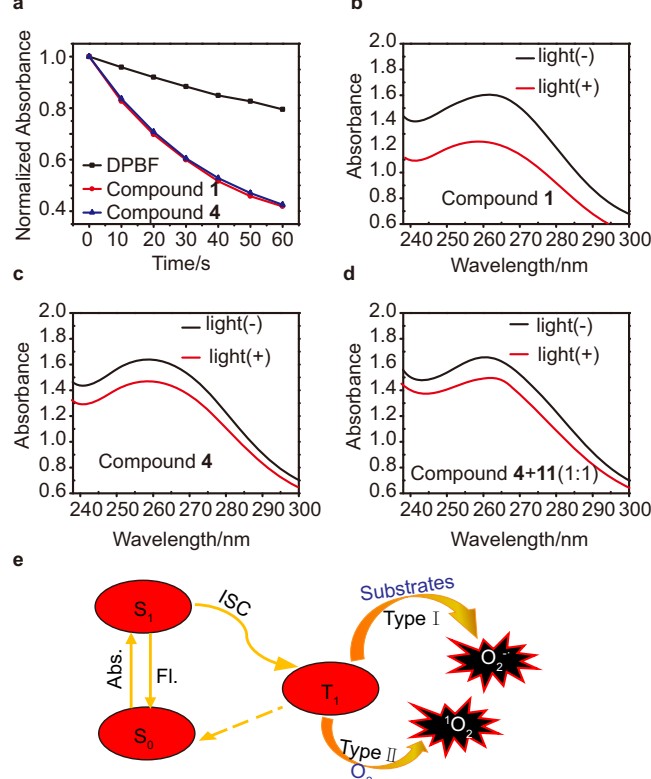

**Fig. 3 ROS generation of compound 1. a** Normalized absorbance of DPBF
(50 μM) at 411 nm in the presence of compound **1** (10 μM) or **4** (10 μM) as
a function of irradiation time (white light: 20 mW/cm$^2$). **b–d** Degradation
of NBT (24 μM) by $O_2^{-\bullet}$ under different treatments ("+" represents with
irradiation, "−" represents without irradiation, white light 20 mW/cm$^2$,
3 min). **e** Schematic illustration of $O_2^{-\bullet}$ and $^1O_2$ generation. Abs. Absorb, Fl
fluorescence, ISC Intersystem Crossing, $S_0$ ground state, $S_1$ singlet state, $T_1$
triplet state.

with the commercially available Protoporphyrin (PpIX, compound **6**), a typical Type II photosensitizer (Fig. 1 and Supplementary Fig. 1). The introduction of biotin unit to the three photosensitizers was originally aimed to provide them the ability of tumor targeting. Compounds **4**, **5**, **6**, and **11** are employed as control compounds to be discussed in the work. The chemical structures of compound **1**, **2**, **3**, **4**, **5**, and **11** were fully characterized by $^1H$ NMR, $^{13}C$ NMR, HRMS, and FTIR (Supplementary Figs. 2–22).

As shown in Fig. 2, compounds **1**, **2**, and **3** exhibited similar absorbance spectra and emission spectra with their precursors **4**, **5**, and **6**, respectively. In particular, **1** is characterized by broader absorption in 400–700 nm, qualifying it as an ideal PS for white-light harvesting (Fig. 2a and Supplementary Fig. 23). Meanwhile, compound **1** emits near-infrared (NIR) fluorescence centered at 750 nm (Fig. 2b). The biotinylation provides the photosensitizers with the ability of targeting tumor cells. Thus, compound **1** holds application prospects for NIR bioimaging and image-guided white-light-induced PDT for malignant tumors.

**PS-Biotin-sensitized ROS generation.** According to our previous works, fluorescein derivatives could be used as ideal candidates for Type II PDT. Therewith, 1,3-diphenylisobenzofuran (DPBF) was employed to appraise the generation of $^1O_2$[39]. As indicated by the DPBF decay curves (Fig. 3a and Supplementary Figs. 24 and 25), irradiation of PS-Biotin or its precursor with white light led to a comparable $^1O_2$ generation, implying that the conjugation of biotin moieties hardly matter to the production of singlet oxygen. In other words, biotinylation has little effect on the energy-transfer process from excited photosensitizers to $O_2$.

However, the biotinylation was found to have an unexpected effect on the electron transfer process involved the Type I PDT. Herein, the generation of $O_2^{-\bullet}$ was measured by the nitrotetrazolium blue chloride (NBT). It is known that NBT can be

specifically reduced by $O_2^{-\bullet}$ to form the insoluble NBT-formazan (Supplementary Fig. 26a) accompanied with the characteristic decrease of the absorbance at 260 nm[40]. According to the results shown in Fig. 3b–d, the production of $O_2^{-\bullet}$ in control groups (compound **4** with irradiation, compound **4** and **11** with irradiation) were less than that in compound **1** with irradiation. Similar results were found in compounds **2** and **3** (Supplementary Fig. 27) indicating that the covalently modified by biotin has a positive effect on the $O_2^{-\bullet}$ generation of photosensitizers. This inference was supported by massive precipitation observed in the irradiation groups for all the three photosensitizers (Supplementary Fig. 26b–d).

Compared with the fluorescein derivatives **4** and **5**, the compound **6**, porphyrin was well known as a classic Type II PS and did not produce obvious $O_2^{-\bullet}$ even under a longer time of irradiation. So, the $O_2^{-\bullet}$ generation ability of compound 3 should be attributed to the introduction of biotin (Supplementary Figs. 26d and 27d–f). Such an unexpected strategy of biotinylation suggested that PS-Biotin could serve as an excellent $O_2^{-\bullet}$ and $^1O_2$ generator at the same time (Fig. 3e). Due to the hypoxia tolerance of Type I and the high reactivity of Type II, PS-Biotin could realize a combinational PDT through Type I and Type II mechanisms. Considering that compound 1 has a stronger ability of $O_2^{-\bullet}$ generation than compound **3**, and matches better with the spectral emission of the white-light source than compound **2**, compound **1** was chosen for demonstration its PDT performance against tumor hypoxia in the following investigation.

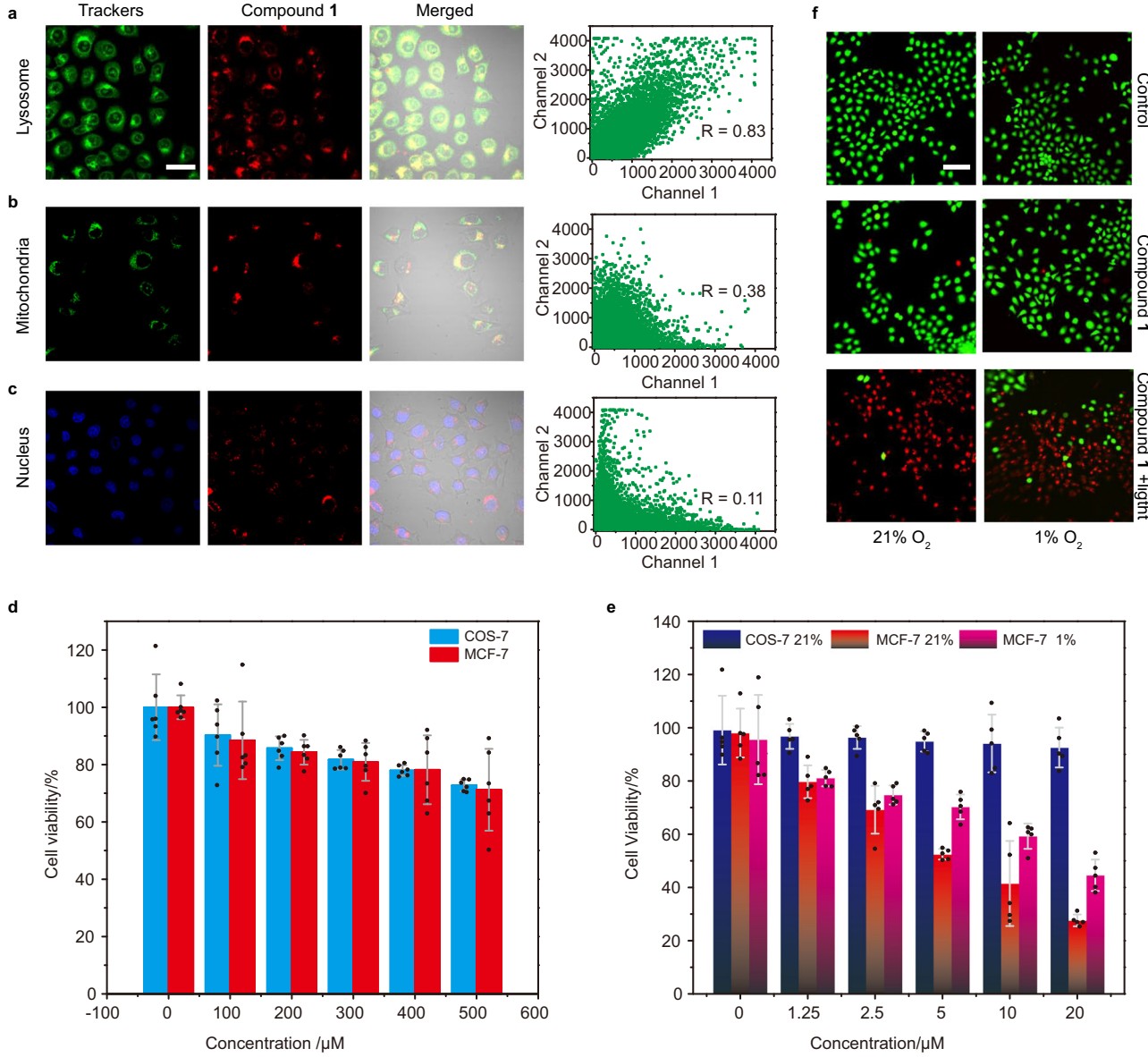

**Fig. 4 Cellular uptake and in vitro PDT effect of compound 1.** Subcellular colocalization images of compound **1** (10 µM) in MCF-7 cells with **a** lysosome-localized tracker LysoTracker Green (1 µM), **b** mitochondria-localized tracker Rhodamine 123 (1 µM), and **c** nuceu-localized tracker Hochest 33324 (1 µM), respectively. **a**–**c** right panel represents the colocalization coefficient of compound **1** and each tracker is 0.83, 0.38, and 0.11, respectively. Scale bar: 30 µm. **d** Dark toxicity of compound **1** on COS-7 cells and MCF-7 cells under normoxia. Data in (**d**) are presented as mean ± s.d. derived from n = 6 independent biological samples. **e** Cell viability of COS-7 cells and MCF-7 cells treated with increasing concentrations of compound **1** after exposure to white-light irradiation for 10 min (white light 20 mW/cm$^2$). Data in (**e**) are presented as mean ± s.d. derived from $n = 5$ independent biological samples. **f** Live/dead cell costaining assays using Calcein-AM and propidium iodide as fluorescence probes under normoxic (21% O$_2$) or hypoxic (1% O$_2$) conditions (green fluorescence for live cells, red fluorescence for dead cells, white light 20 mW/cm$^2$ for 10 min). Scale bar: 100 µm.

**Subcellular colocalization.** Before the PDT experiments, the targeting and imaging capacities of compound **1** and **4** were assessed by confocal laser scanning microscopy (CLSM). Compound **1** was incubated with COS-7 (biotin receptor-negative) and MCF-7 (biotin receptor-positive), respectively. As illustrated in Supplementary Fig. 28, the fluorescence intensity of compound **1** in MCF-7 cells significantly increased over incubation time, while almost no fluorescence was observed in COS-7 cells. As for compound **4** (Supplementary Fig. 29), obvious fluorescence was observed in neither COS-7 cells nor MCF-7 cells. These results indicated that compound **1** could specifically bind to biotin receptors, which were overexpressed on the surface of cancer cells rather than normal cells. These results proved that the

biotinylation can offer the photosensitizer the targeting ability of tumor cells. Then, the intracellular distribution of compound **1** was investigated using commercial organelle-selective trackers. As shown in Fig. 4a, the red signals of compound **1** nicely overlapped with the green fluorescence of Lyso-sensor Green (Pearson correlation coefficient $R = 0.83$). It evidenced that compound **1** was accumulated selectively in the lysosomes. In contrast, only a small amount of compound **1** was distributed in mitochondria and nucleus (Fig. 4b, c).

**In vitro assessment of PDT efficacy.** In order to evaluate the PDT performance of compound **1**, methyl thiazolyltetrazolium

(MTT) assay was carried out. Compound **1** displayed a negligible dark cytotoxic effect to COS-7 cells and MCF-7 cells, suggesting its good biocompatibility in vitro (Fig. 4d). Then, PDT performance was investigated under both normoxia and hypoxia, under white-light (400–800 nm) irradiation of 20 mW/cm². As seen from Fig. 4e, MCF-7 cells were obviously destroyed by **1** ($IC_{50} = 10.3 \mu M$ in MCF-7 cells), while it did not exert influence on COS-7 cell proliferation under normoxia. It was confirmed that the realization of precision PDT is ascribed to photosensitizers based on tumor targeting and selective localization of organelles, which effectively avoid damage to normal cells. Importantly, the PDT efficacy in hypoxic (1% $O_2$) environment is very close to that in normoxia. In detail, ~60% of the cells were killed when treated with 20 μM of compound **1**, indicating that the photocytotoxicity of compound **1** was mainly attributed to the other types of ROS instead of $^1O_2$ under hypoxic condition. The generated ROS should be $O_2^{-\bullet}$ according to the results shown in Fig. 3. Furthermore, the oxygen concentration had minimal effect on cell survival in the dark (Supplementary Fig. 30). Consequently, the conjugation of biotin moieties successfully endowed compound **1** with outstanding cancer selectivity and the capacity of $O_2^{-\bullet}$ generation, which allowing it to achieve targeted phototherapy regardless of oxygen dependence.

The PDT effectiveness of compound **3** and **6** were also assessed. As presented in Supplementary Fig. 31, compound **3** possessed a better photodynamic efficiency both under normoxia and hypoxia than compound **6** because it could produce both $^1O_2$ and $O_2^{-\bullet}$, and the type I mechanism could still function well under depleted oxygen.

The hypoxia-tolerance PDT performance of compound **1** was also confirmed by CLSM experiments. We carried out the live/dead cell staining with Calcein-AM (green fluorescence for live cells) and propidium iodide (PI, red fluorescence for dead cells) to visualize the PDT outcome under both normoxic and hypoxic condition. The CLSM images in Fig. 4f intuitively demonstrated the severe damage to cancer cells by compound **1** exposed to white light under hypoxic condition, and its extensive red PI fluorescence was comparable to that under normal oxygen condition. These results proved that compound **1** could get rid of $O_2$-dependence of traditional Type II PDT.

### The PDT mechanism of compound 1.

To further clarify the particular role of biotinylation in the hypoxia-tolerance PDT, MCF-7 cells treated with compound **1** were stained with the nonspecific ROS probe DCFH-DA[41]. As demonstrated in Fig. 5a–e, upon irradiation, a brightly green fluorescence of DCF was observed, implying the elevated intracellular ROS level by photo-activating compound **1** in MCF-7 cells. The fluorescence intensity was enhanced with the rise of light power from 10 to 20 mW/cm², indicating that the production of ROS was related to the light dose. Gratifyingly, conspicuous green fluorescence of DCF was still detected in MCF-7 cells even in the case of hypoxic environment, highlighting that compound **1** was not seriously susceptible to oxygen depletion. The similar results were also verified by compound **3**. In marked contrast, there was no fluorescence for compound **6** but obvious fluorescence for compound **3** under hypoxia (Supplementary Fig. 32a). It means that the biotinylation provides other ROS but not $^1O_2$ to kill the tumor cells.

In order to further confirm that the generated ROS is $O_2^{-\bullet}$ other than $^1O_2$ under hypoxia, dihydroethidium (DHE), an indicator of $O_2^{-\bullet}$, was chosen to co-stain MCF-7 cells with compound **1** under normoxic and hypoxic conditions[42]. The oxidized product of DHE by $O_2^{-\bullet}$ could intercalate into DNA to emit red fluorescence. As expected, remarkable red fluorescence

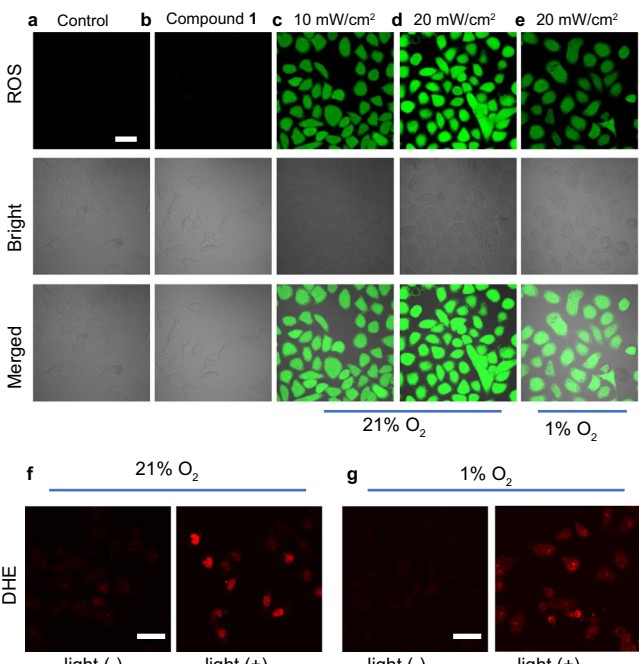

**Fig. 5 In vitro ROS generation of compound 1.** CLSM images of MCF-7 cells stained with DCFH-DA (**a**), compound **1** (10 μM) (**b**), and DCFH-DA + compound **1** with different light doses under (**c**, **d**) normoxic (21% $O_2$) and (**e**) hypoxic (1% $O_2$) conditions. Ex: 488 nm, Em: 500–550 nm. Scale bar: 30 μm. CLSM images of MCF-7 cells stained with compound **1** (10 μM) under normoxic (21% $O_2$, **f**) and hypoxic (1% $O_2$, **g**) conditions using DHE as the intracellular $O_2^{-\bullet}$ fluorescence indicator before and after white-light irradiation ("+" represents with irradiation, "−" represents without irradiation, white light 20 mW/cm², 10 min). Ex: 488 nm, Em: 590–630 nm. Scale bar: 30 μm. Each experiment was repeated three times independently, with similar results.

was still detected in MCF-7 cells even in hypoxia (Fig. 5f, g), meaning that compound **1** can produce $O_2^{-\bullet}$ by irradiation under hypoxia. This type I PDT mechanism involving $O_2^{-\bullet}$ generation would offer PDT more satisfactory efficacy because $O_2^{-\bullet}$ could be catalyzed by intracellular SOD and transformed into other highly cytotoxic radicals (e.g., OH•) through Haber–Weiss reaction and Fenton reaction to realize the reuse of $O_2$, as proved in extensive previous reports[13,18,19,43].

It is noteworthy that the ability of $O_2^{-\bullet}$ generation should be attributed to the biotinylation of the photosensitizers. This inference can also be supported by the following living-cells experiments of compounds **3** and **6**. It is known that photosensitizer **6** is a traditional singlet oxygen generator and could not produce superoxide anions. But **3** can produce $O_2^{-\bullet}$ under white-light irradition in both 21% $O_2$ and 1% $O_2$ conditions (Supplementary Fig. 32b). And these cell-experiments results are consistent with the solution-experiments results of Fig. 3 and Supplementary Fig. 26, indicating that the biotinylation induce the ability of $O_2^{-\bullet}$ generation in the all three photosensitizers.

### PDT efficacy ability on MCTS and in vivo.

As a tissue model, 3D multicellular tumor spheroid (MCTS) was widely used to mimic the conditions in solid tumors such as a hypoxic center and its proliferation gradients[44–46]. MCTS with an average diameter of 800 nm was used to evaluate the PDT efficacy of compound **1**. MCTS was incubated with compound **1** for 4 h,

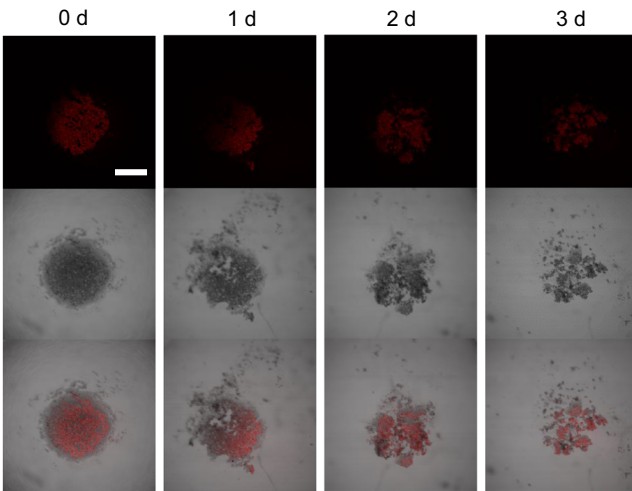

**Fig. 6 Photocytotoxicity of compound 1 to MCF-7 cells 3D MCTS.** The CLSM images of MCTS incubated with compound **1** (10 μM) for 4 h (0 d) and then exposed to white-light irradiation (40 mW/cm$^2$) for 20 min each day (1 d, 2 d, 3 d). The experiment was repeated three times independently, with similar results. Scale bar: 400 μm.

the strong red fluorescence was observed in the entire spheroid, which indicates that the MCTS was completely infiltrated by compound **1** (Supplementary Fig. 33). Upon white-light irradiation (40 mW/cm$^2$, 20 min), the photocytotoxicity of compound **1** was activated, as evidenced by the gradual collapse of MCTS from the superficial part to the inside of the spheroid (Fig. 6). After PDT treatment, MCTS was almost broken down on the 3rd day, whereas the fluorescence of compound **1** was not photo-bleached. This result indicates that compound **1** is a photo-stable white-light photosensitizer (Supplementary Fig. 34). In fact, white-light irradiation alone did not affect the morphology of the MCTS (Supplementary Fig. 35), which proves the white-light PDT is a safe treatment for normal healthy tissues.

The key point to be mentioned that Type II PDT alone is known to have poor efficiency to the core of solid tumors. Satisfactorily, **1** could effectively alleviate hypoxia limitation of conventional PDT. The complete collapse of MCST with hypoxic core was attributed to the Type I and Type II combined therapeutic effect of **1**, which was photo-activated to simultaneously generate $O_2^{-\bullet}$ and $^1O_2$.

The in vivo PDT efficacy was also demonstrated in the mice experiments. Tumor-bearing mice were treated by intravenous injection administration with the photosensitizer **3**. As presented in Supplementary Fig. 36, much better PDT results were achieved in the PDT treatment group than the control groups. In addition, body weights of mice kept increasing in different groups during 18-day treatment, which means that the systemic toxicity of **3** or **6** to mice was negligible.

**The mechanistic explanation of the biotinylation effect.** Based on the above results, we can conclude that the ability of $O_2^{-\bullet}$ generation should be attributed to the biotinylation of the photosensitizers. A mechanistic explanation should be proposed. It is well known that Type I photosensitizers have a lower reduction potential owing to their stronger electron-accepting character[47,48]. We studied the electrochemical properties of compounds **1–6** by cyclic voltammetry with Ferrocene (Fc) as the external standard. As shown in Fig. 7a–c, all the biotinylated three photosensitizers did show lower reductive potential than their counterparts. The anodic shift of the biotinylated photosensitizers

facilitates them to accept electrons, which endow them with the potential to produce more $O_2^{-\bullet}$ by the Type I process. As illustrated in Type I PDT mechanism (Fig. 7d), the triplet PS is transformed into a radical anion by accepting electrons from adjacent substrates and giving external electrons to oxygen to form $O_2^{-\bullet}$[47,49]. Based on the results of theoretical calculation (Fig. 7e), the folded conformation of compound **1** support that more efficient electron transfer can happen between the biotin part and PS part. So, the biotinylation offers an electron-rich substrate in an intramolecular way, which should dramatically benefit to the Type I PDT mechanism.

## Discussion

In summary, we designed and synthesized three PDT photosensitizers **1**, **2**, and **3**, in which the introduction of biotin moiety was aimed to achieve tumor-targeting ability. Unexpectedly, the biotinylation endowed the three photosensitizers the ability of generating $O_2^{-\bullet}$. Considering that the three photosensitizers are different in structures and photophysical properties, the biotinylation can be considered as a potential universal strategy to design PDT photosensitizers which can simultaneously have the functions of tumor targeting and hypoxia tolerance. Photosensitizers designed by the strategy of biotinylation can possess the ability of $O_2^{-\bullet}$ and $^1O_2$ generation at the same time, which would resolve the paradox between traditional Type II PDT and hypoxia environment of solid tumors. Among the three photosensitizers, compound **1** exhibits a broad absorption window (400–700 nm), NIR (750 nm) emission and its combinational Type I/Type II PDT application was demonstrated in this work. We verified that the conjugation of electron-donating biotin moiety to photosensitizers could not only induce the tumor-targeting capability, but also Type I reaction for boosting the production of $O_2^{-\bullet}$ without affecting $^1O_2$ production by our experiments. Besides, 3D multicellular tumor spheroid was effectively disintegrated by compound **1** under white-light irradiation, so as to verify the hypoxia-tolerance PDT performance. In brief, the present work provides a strategy for the practicable design of photosensitizers with synergistic Type I/Type II PDT to relieve the limitation of tumor hypoxia.

## Methods
This research complies with all relevant ethical regulations.

**Materials.** All solvents and reagents were purchased and used as received without further purification. MTT (3-(4,5-dimethyl-2-thiazolyl)-2,5-diphenyl-2H-tetrazolium bromide), LysoTracker Green DND-189, Hoechst 33342 and MitoTracker Green FM, Calcein-AM/propidium iodide (PI) Detection Kit, DHE (Dihydroethidium) were purchased from KeyGEN BioTECH Ltd (Nanjing, China). PpIX (Protoporphyrin) and NBT (nitrotetrazolium blue chloride) were purchased from Aladdin Industrial Corporation (Shanghai, China). All other reagents were purchased from Energy Chemical (Shanghai, China). The purity of all reagents purchased is above 95%. Silica gel (200–300 mesh) was used for flash-column chromatography.

**Computational details.** All calculations were performed by the Gaussian 09 software package. The molecular geometries were optimized by dispersion-corrected density function theory (DFT-D3) at B3LYP level of theory with 6–31 G(d) basis set. The solvent effect (water) was taken into account using the PCM model to simulate the electrostatic environment in an aqueous solution. Cartesian coordinates of optimized compound **1** were provided in Supplementary Table 1.

**$^1O_2$ detection.** The $^1O_2$ generation of compounds **1** and **4** in ethanol was calculated by using DPBF. The absorbance decrease of DPBF at 411 nm was recorded for different durations of white-light irradiation (20 mW/cm$^2$) to obtain the decomposition rate of the photosensitizing process. And during the measure, the initial value of absorption of **1** and **4** was consistent at 475 nm.

The same operation to compounds **2**, **5**, **3**, and **6**.

**$O_2^{-\bullet}$ detection.** The solution of 2.4 mM Nitroblue Tetrazolium (NBT) was prepared with phosphate-buffered saline (PBS, 10 mM). 30 μL NBT was added to 3 mL PBS solutions of **1** (10 μM) and **4** (10 μM) to a final concentration of 24 μM,

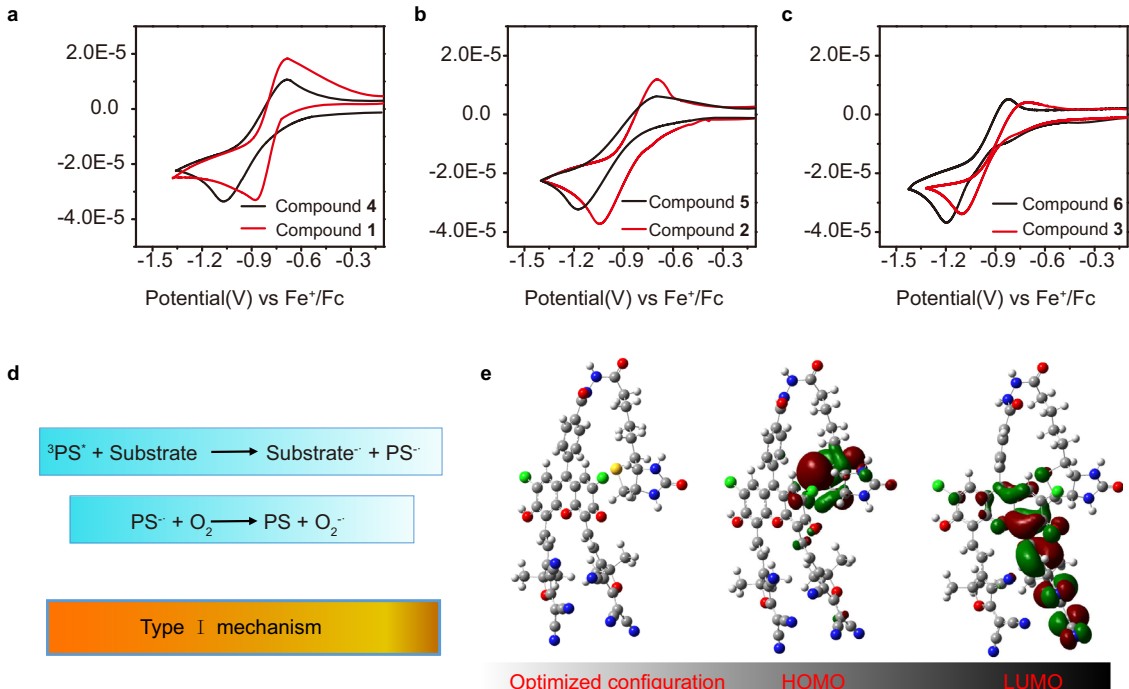

**Fig. 7 The mechanistic explanation of the biotinylation effect.** Cyclic voltammograms of **a** compound **1**, **4**, **b 2**, **5**, and **c 3**, **6** in DMF with 0.1 M (n-Bu)$_4$N$^+$PF$_6^-$ as a supporting electrolyte, glassy carbon as a working electrode, Ag/AgCl as a reference electrode, Pt wire as a counter electrode, and a scan rate of 10 mVs$^{-1}$. Fc/Fc$^+$ was used as an external reference. **d** Photochemical reactions during Type I mechanism ($^3$PS$^*$ represents the triple excited state of photosensitizer, PS$^{-•}$ represents the anion radicals of photosensitizer, $^3$PS$^*$ is transformed into a radical anion by accepting electrons from adjacent substrates and giving external electrons to oxygen to form O$_2^{-•}$). **e** Optimized triplet state structure, HOMO and LUMO of compound **1** (DFT, B3LYP/6-31G (**d**) calculated frontier orbitals relevant to the triplet state).

respectively. The solutions were irradiated for 3 min with white light (20 mWcm$^{-2}$). NBT could react with O$_2^{-•}$ to form a dark insoluble precipitate.

The same operation to compounds **2**, **5**, **3**, and **6**.

**Photo-stability of DCF-TFM-Biotin.** The concentrated DMSO solution of compound **1** (1 mM) was diluted with RPMI-1640 cell culture medium containing 10% fetal bovine serum (FBS) into 10 μM. The sample tube was exposed to a white light (400–800 nm, 40 mW/cm$^2$). Fluorescence intensity values of 733 nm for compound **1** was recorded after every 20 min. The excitation wavelength was 515 nm.

**Cyclic voltammetry measurement.** The cyclic voltammetry measurement of compounds **1–6** was conducted in DMF with 0.1 M (n-bu)$_4$N$^+$PF$_{6-}$ as a supporting electrolyte, glassy carbon as a work electrode, Ag/AgCl as a reference electrode, Pt wire as a counter electrode, and a scan rate of 10 mVs$^{-1}$. Fc/Fc$^+$ was used as an external reference.

**Cell culture.** COS-7 cells were cultured in Dulbecco's Modified Eagle Medium (DMEM; GIBCO) containing 10% fetal bovine serum (FBS; GIBCO) and 1% antibiotics (80 U mL$^{-1}$ penicillin and 0.08 mg mL$^{-1}$ streptomycin; GIBCO). MCF-7 cells were cultured in RPMI-1640 cell culture medium (GIBCO) containing 10% fetal bovine serum (FBS; GIBCO) and 1% antibiotics (80 U mL$^{-1}$ penicillin and 0.08 mg mL$^{-1}$ streptomycin; GIBCO). Cultured cells were incubated in an atmosphere of 5% CO$_2$ at 37 °C.

**Confocal fluorescence imaging of cells.** COS-7 cells and MCF-7 cells were planted onto 35-mm confocal dishes at a density of 1 × 10$^5$ cells. After incubation for 24 h, the medium was removed and washed with PBS twice. Fresh medium containing compound **1** (10 μM) was added and incubated for different times at 37 °C. COS-7 cells and MCF-7 cells were co-cultivated with compound **1** (10 μM) for 4 h at 37 °C. After removal of the medium and PBS washing for three times and the addition of fresh medium, cell image was conducted using the confocal laser scanning microscopy (CLSM). The excitation wavelength was 635 nm. Collection wavelength was from 690 nm to 790 nm for **1** and **4**.

**Intracellular ROS detection.** 2′, 7′-dichlorodihydrofluorescein diacetate (DCFH-DA) was employed as the intracellular ROS indicator, which can be converted to DCF and emits bright green fluorescence in the presence of ROS. MCF-7 cells were planted onto 35 mm confocal dishes at a density of 1×10$^5$ cells and cultured for

24 h at 37 °C under 5% CO$_2$. The cells were then incubated with 10 μM compound **1** for 4 h. After rinse with PBS, the cells were incubated with 1 μM DCFH-DA for another 30 min. The cells were washed with PBS and exposed to irradiation for 10 min with a LED lamp (400–800 nm, 20 mW/cm$^2$). After irradiation, confocal fluorescence imaging was used to observe the intracellular ROS level. The excitation wavelength for DCF was 488 nm and emission wavelength collected from 500 nm to 550 nm for DCF. In order to simulate hypoxic environment (1% O$_2$), Anaero Pack-Anaero and Anaero Pack-Micro Aero (Mitsubishi Gas Chemical Company, Japan) were used. MCF-7 cells were planted onto 35 mm confocal dishes at a density of 1 × 10$^5$ cells and cultured for 16 h under normoxic condition, and then the cells were incubated for another 8 h at 37 °C under hypoxic. Other operations were same to that in normoxic condition.

**Intracellular O$_2^{-•}$ detection.** The detection of O$_2^{-•}$ was performed using the similar procedure described for the detection of ROS, except that DHE (10 μM) was used as the O$_2^{-•}$-specific probe. The red fluorescence signal of cells was collected by CLSM.

**Subcellular colocalization assay.** MCF-7 cells were planted onto 35 mm confocal dishes at a density of 1 × 10$^5$ cells and incubated with 10 μM compound **1** for 4 h at 37 °C under 5% CO$_2$. The cells were further stained by 1 μM LysoSensorTM Green DND-189 for 30 min or 1 μM Hoechst 33342 or 1 μM Rho 123 for 10 min. Cells were then visualized with laser confocal microscopy. The excitation wavelength for compound **1** was 635 nm, while the excitation wavelength for LysoSensorTM Green DND-189 and Rho 123 was 488 nm, for Hoechst 33342 was 405 nm. The emission wavelength was collected from 690 to 790 nm for compound **1**, 500–550 nm for LysoSensorTM Green DND-189, and Rho 123,440 to 480 nm for Hoechst 33342.

**Calcein-AM/PI staining of MCF-7 cells.** MCF-7 cells were planted onto 35 mm confocal dishes at a density of 1 × 10$^5$ cells for 24 h at 37 °C under 5% CO$_2$. MCF-7 cells incubated with different following treatments: group 1, untreated; group 2, incubated with 10 μM compound **1** for 4 h without light irradiation; group 3, incubated with 10 μM compound **1** for 4 h followed by 400–800 nm LED light at a light dose of 20 mW/cm$^2$ for 10 min. Before imaging, each group was stained with 2 μM Calcein-AM and 8 μM PI for 30 min. Then the fluorescence images of Calcein-AM/PI within MCF-7 cells were detected using confocal microscopy with the excitation wavelength of 488 nm, capture emission region from 500 nm to 550 for green channel, 600–640 nm for red channel.

**In vitro PDT cytotoxicity assay**. The MCF-7 or COS-7 cells were planted in 96-well plate (5000 per well) for 16 h, and another 8 h under normoxic (21% $O_2$) or hypoxic (1% $O_2$) atmosphere. After 24 h, the compound **1** at different concentrations was added and continued to incubate 4 h under normoxic (21% $O_2$) or hypoxic (1% $O_2$) atmosphere. After that, the cell culture media was replaced with 100 μL fresh medium. Subsequently, the cells were irradiated upon white light for 10 min (20 mW/cm²). After irradiation, the cells were again incubated for 24 h. Then, 20 μL MTT solution (5 mg /mL) in pH 7.4 PBS was added to each well. After 4 h of incubation, the medium was removed carefully, and 150 μL DMSO was added to each well to dissolve the produced blue formazan. The absorbance value of each well was recorded with a microplate reader at 540 nm. The cell viability rate was calculated by the following equation:

$$\text{Cell viability}(\%) = \frac{OD_{PS} - OD_{Blankcontrol}}{OD_{Control} - OD_{Blankcontrol}} \times 100\% \tag{1}$$

For dark toxicity measurement of compound **1**, no light irradiation was applied to this experiment, and other steps were the same.

**3D multicellular tumor spheroid (MCTS) model of MCF-7 cells**. Multicellular tumor spheroids (MCTSs) from MCF-7 cells were obtained using hanging drop technique. 18 μL cell suspension was seeded on the lid, and then inverted onto the confocal dish and cultured for 3–6 days, consequently aggregating into tumor spheroids.

**In vitro penetration**. When the size of MCTSs reached about 800 μm in diameter, MTCSs were treated with compound **1** (10 μM) for 4 h. Optical sections of MTCSs were imaged under a CLSM from top to bottom with 25 μm per section.

**PDT efficacy ability on MCTS**. MCTSs were incubated with compound **1** (10 μM) for 4 h. The spheroids were irradiated with a 40 mW/cm² white light for 20 min. PDT experiments of MCTS last for 3d. The control group was only irradiated for 20 min day⁻¹.

**Animals and tumor model**. All animal experiments involved in this study have been approved by the local research ethics review board of the Animal Ethics Committee of the Dalian University of Technology (Certificate number//Ethics approval no. is 2018–043). 6-week-old female BALB/c mice were collected from the Laboratory Animal Center of Dalian Medical University to establish a breast cancer mouse model. In brief, $1 \times 10^6$ 4T1 cells were subcutaneously injected into the right hind leg to establish 4T1 tumor-bearing BALB/c mice. The tumor volume of 4T1-bearing mice was calculated as $A=a*b^2/2$ (*a*: length; *b*: width). Mice were treated for photodynamic therapy after the tumor volume was about 100 mm³. Mice were kept in transparent, nontoxic plastic boxes with stainless-steel wire cages. Laminar flow cabinet and separate ventilation cages were used to purify the air in the cages. Mice were reared at 23–26 °C, with a relative humidity of 50–55% and alternating light and dark for 12 h/12 h. Mice were fed 2–3 times a week, and drinking water bottles were cleaned and disinfected 3–4 times a week. Padding were replaced twice a week.

**In vivo PDT evaluation**. To evaluate the in vivo PDT efficacy of compound **3** or **5**, all mice were divided into five groups and performed with the following different treatments: group 1, Saline; group 2, only compound **5** injection; group 3, only compound **3** injection; group 4, compound **5** injection and irradiation; group 5, compound **3** injection and irradiation. Each group contained three mice, and compound **3** or **5** (100 μmol/mL, 100 μL) was intravenously injected. After 6 h post-injection, tumor region was irradiated with 630 red LED light at a power density of 50 mW/cm² for 30 min. The tumor volume of all mice was measured using a vernier caliper for 18 days after different treatments. Then, the greatest longitudinal diameter (length) and the greatest transverse diameter (width) were used to calculate the tumor volume in vivo photodynamic therapy experiments. Isoflurane was used as an anesthetic in vivo photodynamic therapy experiments. After 18 days of treatment, the mice were killed by spinal dislocation.

**Syntheses of compounds**

*4-(2,7-dichloro-6-hydroxy-3-oxo-3H-xanthen-9-yl) benzoic acid (7)*. 4-formylbenzoic acid (624 mg, 4.16 mmol) and 4-chlorobenzene-1,3-diol (1.2 g, 8.32 mmol) were added to a 100 mL round bottom flask. 13 mL of methanesulfonic acid was added and magnetically stirred at 110 °C for 16 h. The reaction mixture was allowed to cool to room temperature. The reaction solution was then added dropwise to 500 mL of cold water, followed by filtration and washing of the filter cake three times with deionized water (50 mL). After drying, the crude product **7** (1.59 g) was obtained and used directly in the next reaction step.

*4-(2,7-dichloro-4,5-diformyl-6-hydroxy-3-oxo-3H-xanthen-9-yl) benzoic acid (8)*. Compound **7** (400 mg, 1.0 mmol) and hexamethylenetetramine (1.0 g, 5.0 mmol) were dissolved in trifluoroacetic acid (15 mL) and magnetically stirred at 90 °C overnight. Then, cooled to room temperature. 2 N hydrochloric acid was added

with stirring until the orange solid did not precipitate and the cake was washed with deionized water until the filtrate became colorless. After drying, the crude product **8** (429.2 mg) was obtained and used directly in subsequent reactions.

*2-(3-cyano-4,5,5-trimethylfuran-2(5H)-ylidene) malononitrile (9)[50]*. 3-hydroxy-3-methylbutan-2-one (1.02 g, 10.0 mmol), malononitrile (1.98 g, 30.0 mmol) and magnesium ethoxide (1.25 g, 11.0 mmol) were dissolved in ethanol and magnetically stirred at 60 °C overnight. Then the solvent was evaporated in vacuum. After that, 50 mL of dichloromethane was added to the reaction flask, sonicated, extracted and the filter cake was washed with dichloromethane (10 mL × 3). The filtrate was evaporated in vacuum to obtain a yellowish-brown crude product **9** (1.91 g) and used directly in subsequent reactions.

*2-(2,6-dimethyl-4H-pyran-4-ylidene) malononitrile (10)[51]*. 2,6-dimethyl-4*H*-pyran-4-one (0.62 g, 5.0 mmol) and malononitrile (0.33 g, 5.0 mmol) were added to a 100 mL round bottom flask. 10 mL of acetic anhydride was added to the mixture and stirred at 140 °C for 6 h. The heating and stirring were stopped and the reaction solution was stood overnight, and filtered. The filter cake was washed with n-hexane (15 mL ×3) and dried to give a brown crude product **10** (0.832 g).

*4-(2,7-dichloro-4,5-bis((E)-2-(4-cyano-5-(dicyanomethylene)-2,2-dimethyl-2,5-dihydrofuran-3-yl)vinyl)-6-hydroxy-3-oxo-3H-xanthen-9-yl) benzoic acid (4)*. Compound **8** (228 mg, 0.5 mmol) were dissolved in methanol (25 mL). Compound **9** (251 mg, 1.25 mmol) was added and magnetically stirred at 60 °C overnight. The reaction mixture was allowed to cool to 25 °C. Then the solvent was evaporated in vacuum. The crude product was purified by column chromatography (methanol / dichloromethane = 1:100–1:8). Compound **4** was obtained as black brown solid (100 mg, 24.4% yield). ¹H NMR (400 MHz, DMSO-d6) δ 9.04 (d, *J* = 15.2 Hz, 2 H), 8.21 (d, *J* = 7.8 Hz, 2 H), 8.07 (d, *J* = 15.2 Hz, 2 H), 7.59 (d, *J* = 7.8 Hz 2 H), 7.07 (s, 2 H), 1.67 (s, 12 H); ¹³C NMR (126 MHz, DMSO-d6) δ 178.7, 177.8, 173.2, 155.8, 152.1, 139.6, 130.2, 129.7, 129.4, 127.5, 113.6, 113.0, 112.9, 112.1, 110.2, 110.0, 98.9, 92.2, 52.2, 24.7. HRMS (*m/z*): [M–H]⁻, found, 817.1010; calcd. for $C_{44}H_{23}Cl_2N_6O_7$, 817.1005.

*4-(2,7-dichloro-4,5-bis((E)-2-(4-(dicyanomethylene)-6-methyl-4H-pyran-2-yl)vinyl)-6-hydroxy-3-oxo-3H-xanthen-9-yl) benzoic acid (5)*. Compound **8** (114 mg, 0.25 mmol) and **10** (100 mg, 0.5 mmol) were dissolved in methanol (15 mL). Piperidine (100 μL) was added and magnetically stirred at 70 °C overnight. The reaction mixture was allowed to cool to 25 °C. Then the solvent was evaporated in vacuum. The crude product was purified by column chromatography (methanol / dichloromethane = 1: 100 to 1:8). Compound **5** was obtained as black solid (39.7 mg, 20.8% yield). ¹H NMR (400 MHz, DMSO-d6) δ 8.19 (d, *J* = 7.7 Hz, 2H), 7.98 (s, 2H), 7.71 (s, 1H), 7.67 (s, 1H), 7.61 (d, *J* = 7.8 Hz, 2H), 6.93 (s, 2H), 6.60 (s, 2H), 6.41 (s, 2H), 2.33 (s, 6H). ¹³C NMR (126 MHz, DMSO-d6) δ 172.7, 163.9, 162.0, 156.4, 154.9, 130.4, 130.0, 129.9, 129.3, 127.2, 118.2, 115.9, 110.7, 109.0, 106.0, 105.3, 19.8. HRMS (*m/z*): [M–H]⁻, found,763.08055; calcd. for $C_{42}H_{21}Cl_2N_4O_7$, 763.0787.

*5-(2-oxohexahydro-1H-thieno[3,4-d]imidazol-4-yl) pentanehydrazide (11)*. Compound **11** was prepared by a modification of previous methods[52]. SOCl₂ (0.30 mL, 4.0 mmol) was added to the suspension of biotin (300 mg, 1.23 mmol) in MeOH (3 mL), and the solution was stirred overnight at room temperature to give a clear solution. After evaporation of the solvent and excess SOCl₂ under reduced pressure, biotin methyl ester (296 mg) was obtained and used directly for the next reaction.

Biotin methyl ester (296 mg, 1.14 mmol) was dispersed in MeOH (2.5 mL), and hydrazine (0.48 mL, 10 mmol) was added with stirring. The mixture was refluxed for 16 h at 70 °C. Cooled to room temperature, excessive methanol was added for washing, extraction and drying to give Biotin–NH–NH2 (279.4 mg, 95% yield) as a white solid: ¹H NMR (400 MHz, D₂O) δ4.55–4.52 (m, 1 H), 4.37-4.34 (m, 1 H), 3.29–3.24 (m, 1 H), 2.95–2.90 (dd, *J* = 13.0, 5.0 Hz, 1 H), 2.71 (d, *J* = 13.0 Hz, 1 H), 2.15 (*t*, *J* = 7.3 Hz, 2 H),1.70–1.46 (m, 4 H), 1.39–1.29 (m, 2 H); ¹³C NMR (126 MHz, D₂O) δ 24.8, 27.6, 27.8, 33.4, 39.7, 55.3, 60.2, 62.0, 165.4, 175.6. HRMS (*m/z*): [M + H]⁺, found, 259.12229; calcd. for $C_{10}H_{19}N_4O_2S$, 259.1229.

*4-(2,7-dichloro-4,5-bis((E)-2-(4-cyano-5-(dicyanomethylene)-2,2-dimethyl-2,5-dihydrofuran-3-yl)vinyl)-6-hydroxy-3-oxo-3H-xanthen-9-yl)-N'-(5-(2-oxohexahydro-1H-thieno[3,4-d]imidazol-4-yl)pentanoyl)benzohydrazide (1)*. Compound **4** (50 mg, 0.06 mmol) was dissolved in anhydrous DMF (8 mL), then stirred in ice bath at $N_2$ atmosphere. EDCI (60 mg, 0.31 mmol), 1-Hydroxybenzotriazole (HOBt, 42.5 mg, 0.31 mmol) and N, N- Diisopropylethylamine (DIEA, 31 μL, 0.18 mmol) was added into the system and stirred for 1.5 h, and then compound **11** (32.5 mg, 0.13 mmol) was added. After stirring at 0 °C for 1 h, then the reaction complex was stirred for 12 h at room temperature. The solvent was evaporated under reduced pressure. The reaction mixture was purified by flash-column chromatography (methanol / dichloromethane = 1:100–1:10) to obtained compound **1** (12.7 mg, 20% yield) as black solid. ¹H NMR (400 MHz, DMSO-d6) δ 10.53 (s, 1H), 9.97 (s, 1H), 9.04 (d, *J* = 15.2 Hz, 2H), 8.13 (d, *J* = 7.9 Hz, 2H), 8.07 (d, *J* = 15.2 Hz, 2H), 7.64 (d, *J* = 8.0 Hz, 2H), 7.07 (s, 2H), 4.33 (dd, *J* = 7.8, 5.0 Hz, 1H), 4.20 – 4.15 (m, 1H), 3.17 (d, *J* = 4.6 Hz, 1H), 3.01 (dt, *J* = 20.4, 7.0 Hz, 2H), 2.84 (d, *J* = 5.0 Hz, 1H), 2.73

(s, 2H), 2.24 (t, J = 7.3 Hz, 2H), 1.67 (s, 12H), 1.04 (d, J = 6.1 Hz, 2H), 0.98 (t, J = 7.2 Hz, 1H). $^{13}$C NMR (126 MHz, DMSO-d6) δ 178.7, 177.8, 173.2, 171.5, 165.0, 155.8, 152.0, 139.6, 130.2, 129.3, 128.0, 127.5, 113.7, 113.0, 112.9, 112.1, 110.2, 110.0, 98.9, 92.2, 61.0, 59.2, 55.4, 52.2, 41.9, 34.0, 33.1, 28.1, 25.2, 24.7. HRMS (m/z): [M–H]⁻, found, 1057.2040; calcd. for $C_{54}H_{40}Cl_2N_{10}O_8S$, 1057.2050.

*4-(2,7-dichloro-4,5-bis((E)-2-(4-(dicyanomethylene)-6-methyl-4H-pyran-2-yl)vinyl)-6-hydroxy-3-oxo-3H-xanthen-9-yl)-N'-(5-(2-oxohexahydro-1H-thieno[3,4-d]imidazol-4-yl)pentanoyl)benzohydrazide* (**2**). It's the same method as the synthesis of compound **1**. The reaction mixture was purified by flash-column chromatography (methanol/dichloromethane = 1:50–1:8) to obtained compound **2** (10.3 mg, 15.2% yield) as black solid. $^1$H NMR (400 MHz, DMSO-d6) δ 10.52 (s, 1H), 9.96 (s, 1H), 8.10 (d, J = 7.9 Hz, 2H), 8.01–7.85 (m, 2H), 7.82–7.65 (m, 3H), 7.64 (s, 1H), 6.91 (s, 2H), 6.59 (s, 2H), 6.37 (s, 2H), 4.33 (t, J = 6.5 Hz, 1H), 4.18 (d, J = 5.1 Hz, 1H), 3.16 (d, J = 7.6 Hz, 1H), 2.84 (q, J = 6.8, 6.1 Hz, 2H), 2.31 (s, 6H), 2.09–1.87 (m, 2H), 1.73–1.51 (m, 4H), 0.89–0.76 (m, 2H). $^{13}$C NMR (126 MHz, DMSO-d6) δ 172.2, 171.5,165.2, 163.3, 162.7, 161.4, 161.2, 158.4, 155.9, 154.3, 150.2, 133.5, 129.8, 128.9, 127.7, 126.7, 117.6, 115.4, 115.1, 110.1, 108.6, 105.5, 104.8, 61.0, 59.2, 55.4, 54.5, 42.0, 36.2, 34.1, 28.0, 25.2, 19.3. HRMS (m/z): [M–H]⁻, found, 1003.1838; calcd. for $C_{52}H_{38}Cl_2N_8O_8S$, 1003.1832.

*3-(2,8,12,17-tetramethyl-3-(3-oxo-3-(2-(5-(2-oxohexahydro-1H-thieno[3,4-d]imidazol-4-yl)pentanoyl)hydrazinyl)propyl)-13,18-divinylporphyrin-7-yl)propanoic acid* (**3**). Compound **6** (50 mg, 0.089 mmol) was dissolved in anhydrous DMF (8 mL), then stirred in ice bath at N$_2$ atmosphere. EDCI (17 mg, 0.089 mmol), HOBt (12 mg, 0.089 mmol) and DIEA (15.5 μL, 0.089 mmol) was added into the system and stirred for 1.5 h, and then compound **11** (20 mg, 0.077 mmol) was added dropwise. After stirring at 0 °C for 1 h, then the reaction complex was stirred for 12 h at room temperature. The solvent was evaporated under reduced pressure. The reaction mixture was purified by flash-column chromatography (methanol/dichloromethane = 1:50–1:8) to obtained crude products, then dissolved with hot methanol, precipitated out, and washed the residue with methanol (3 × 10 mL). After drying, obtained compound **3** (12.8 mg, 17.9% yield) as black solid. $^1$H NMR (400 MHz, DMSO-d6) δ 10.14–9.98 (m, 6H), 8.44 (s, 2H), 6.45–6.35 (m, 2H), 6.21 (d, J = 11.8 Hz, 2H), 4.29 (s, 5H), 4.10 (s, 1H), 3.68 (s, 6H), 3.58 (s, 6H), 3.16–3.10 (m, 7H), 2.09 (t, J = 7.7 Hz, 2H), 1.51 (s, 4H), 1.31 (s, 2H), −4.16 (s, 2H). $^{13}$C NMR (126 MHz, DMSO-d6) δ170.9, 170.3, 162.7, 130.0, 121.0, 97.1, 61.0, 59.1, 55.3, 39.9, 37.0, 36.1, 32.9, 28.0, 25.0, 21.5, 12.5, 11.3. HRMS (m/z): [M–H]⁻, found, 801.35614; calcd. for $C_{44}H_{49}N_8O_5S$, 801.3547.

**Instrument and software**. $^1$H NMR was performed on a 400-MHz Varian using Me$_4$Si as the internal standard. $^{13}$C NMR was performed on a 100-MHz Varian. Mass spectrometry was recorded on a HP 1100 LC/MS manufactured by American Agilent. Absorption spectra were measured from a UV-Vis spectrophotometer (Agilent Tech). Fluorescence spectroscopy were recorded on a fluorometer (Agilent Tech). Cyclic voltammetry measurement was conducted by using CHI760E Electrochemical workstation (Shang Hai Chen Hua). Fluorescence imaging was performed by using an OLYMPUS FV-1000 inverted fluorescence microscope. The FTIR spectra were obtained from Fourier Transform Infrared Spectrometer (Vertex 70 v). OriginPro 9, FV10-ASW 3.0 Viewer, and Image J bundled with 64-bit Java 1.8.0_172 were used for data processing.

**Reporting summary**. Further information on research design is available in the Nature Research Reporting Summary linked to this article.

## Data availability
The experimental data generated in this study are available within the article, Supplementary Information, and Source data. Source data are provided with this paper.

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

## Acknowledgements

This work was supported financially by the National Natural Science Foundation of China (21877011), the Fundamental Research Funds for the Central Universities (DUT20YG119), and the Talent Fund of Shandong Collaborative Innovation Center of Eco-Chemical Engineering (XTCXYX03).

## Author contributions

F.L.S. designed and guided the overall research project. J.A. as the first author designed the experiments and wrote the manuscript. S.L.T. and G.B.H. assisted the synthesis. W.L.C. and M.M.C. involved cellular and mice experiments. J.T.S., Z.L.L., X.J.P., and W.H.Z. provided intellectual input and revised the manuscript.

## Competing interests

The authors declare no competing interests.
