## [Peer review file · Nature Communications]

REVIEWER COMMENTS

Reviewer #1 (Remarks to the Author):

This manuscript submitted to Nature Communications by Jing An and co. workers describes the synthesis of three photosensitizers bearing biotin groups which is specific target molecule for tumor cells. The in vitro photodynamic activities of these photosensitizers were tested to biotin receptor-positive MCF-7 and biotin receptor-negative COS-7 tumor cells for comparison. This study looks a routine work because many photosensitizers are known in the literature and in vitro activities of these photosensitizers were also explained. However, the combination of the singlet oxygen and ROS generation behavior of the synthesized photosensitizers could be interesting and increase the impact of the work. But the most limitation of this works is the using white light as a light source. One of the advantage of the PDT is the used long wavelength (such as NIR region) of the light because the patients do not interact with sun light after treatment of white light activated photosensitizers because these photosensitizers can be activated by sunlight. According to my opinion, this study can be evaluated for publication after reconsidering the shortcomings mentioned above and the comments given below.

Some comments

1. What is the used excitation wavelength of the emission spectra given in Figure 1.
2. a, b, c and d should be labelled in Figure 5.
3. Lines 220 and 221: "respectively" should be removed.
4. Line 246: in "10⁵" 5 should be superscript.
5. Line 256: "2% O₂" should be "1% O₂".
6. What is the reaction temperature for the synthesis of DCF-TFM?
7. Totally 22 H were given in the 1H NMR data of DCF-TFM but this compound contains 24 H.
8. I calculated the molecular weight of the compound DCF-TFM as 819.60. It should be checked.
9. Totally 13 H were given in the 1H NMR data of Biotin-NH₂-NH₂ but this compound contains 18 H.
10. Totally 37 H were given in the 1H NMR data of DCF-TFM-Biotin but this compound contains 40 H.
11. I calculated the molecular weight of the compound DCF-TFM-Biotin as 1059.93. It should be checked.
12. Totally 32 H were given in the 1H NMR data of FL-Biotin but this compound contains 38 H.
13. I calculated the molecular weight of the compound FL-Biotin as 1005.88. It should be checked.

14. Figure S10 did not cite in the manuscript.

15. ¹³C NMR spectra of the compounds FL-Biotin and PpIX-Biotin should be supplied.

Reviewer #2 (Remarks to the Author):

The authors reported three new photosensitizers for PDT studies. The biotination moiety was incorporated in these photosensitizers to enhance the tumor cell selectivity. As expected, the authors noticed that the newly designed sensitizers did show selectivity towards tumor cells over the normal cells. In addition, they found the biotination induced the generation of superoxide radical (O₂⁻), which eventually helps to increase the PDT effect in even hypoxia environments. This effect was attributed to the combination of type I and II mechanisms. Among the three photosensitizers, the DCF-TFM-Biotin was found to be ideal for practical applications. The paper is well written, and sufficient evidence was provided for every conclusion. Therefore, I believe this paper is suitable for publication in Nature Communications.

Minor comments:

1. How the superoxide radicals were produced when there is no oxygen around? I mean, under hypoxia conditions the oxygen levels are very low, so how can produce such large quantities of superoxide radicals?
2. To me it looks like we need O₂ for both types of I and II mechanisms. Is there another pathway for the production of radical (SOD) in Type I mechanism?
3. How the authors avoided double biotination of PpIX?
4. Must show the entire NMR range (Figure S29), from -4 ppm to 10 ppm. Must include inner NH peaks.

Reviewer #3 (Remarks to the Author):

The current paper is part of an ongoing research carried out by the authors on the development of organic photosensitizers, and their application in photodynamic therapy (PDT). More specifically, this article is directed toward the development of organic photosensitizers for PDT type I, in order to overcome hypoxia considered as the Achilles heel of PDT. The authors' basic finding, is that the

biotination of 3 organic photosensitizers increases their ability to generate superoxide anion $O_2^{\bullet-}$, making them interesting candidates for type 1 PDT. Although this finding is quite interesting and in itself supports the publication of the article, there are some points that authors should address before accepting this paper in this journal or elsewhere:

- Although the reported effect of biotination was unexpected, it nevertheless has precedents. Surprisingly, one of the most representative precedent was reported by some of the authors. Indeed, in a full paper published recently in JACS, Peng's group reported that the biotination of ENBS afforded ENBS-B a photosensitizer with a better capacity for $O_2^{\bullet-}$ generation. This article (reference 30), inadequately quoted by the authors, should be cited in the introductory part and the structure of ENBS-B should also be given.

- Apart from Peng's article, the biotination of other photosensitizers has already been reported in the literature, and should be cited (New J. Chem. 2020, 44, 3392-3401; J.Mater.Chem.B, 2019,7,65-79; Chem. Commun., 2019, 55, 10972-10975; J.

Photochem. Photobiol.,B, 2019, 190, 1-7)

- On the other hand, the authors claim that PDT type I based on organic photosensitizers is very rare. Being true some few years ago, the panorama is drastically changing as shown by the increasing number of works reports on this subject. Most of these advances are included in the recently released review by Chen et al. "Type I Photosensitizers Revitalizing Photodynamic Oncotherapy. Small 2021, 17, 2006742". - In the cell selectivity studies, the authors used MCF7 breast cancer cells and kidney Cos-7 cells. Would it not be more accurate to use in this study non-malignant breast cells MCF10 usually used as a reliable model for normal mammary cells?

- The authors must compare the efficiency of their system with the commercially available, clinically-used photosensitizer chlorin e6

- The authors must carry out in vivo studies in order to show the effectiveness of their system in terms of activity, bioavailability and targeting.

- The authors must also conduct mechanistic studies to rationalize the observed effect.

Reply to reviewers

Reviewer 1

Comments: This manuscript submitted to Nature Communications by Jing An and co. workers describes the synthesis of three photosensitizers bearing biotin groups which is specific target molecule for tumor cells. The in vitro photodynamic activities of these photosensitizers were tested to biotin receptor-positive MCF-7 and biotin receptor-negative COS-7 tumor cells for comparison. This study looks a routine work because many photosensitizers are known in the literature and in vitro activities of these photosensitizers were also explained. However, the combination of the singlet oxygen and ROS generation behavior of the synthesized photosensitizers could be interesting and increase the impact of the work. But the most limitation of this works is the using white light as a light source. One of the advantage of the PDT is the used long wavelength (such as NIR region) of the light because the patients do not interact with sun light after treatment of white light activated photosensitizers because these photosensitizers can be activated by sunlight. According to my opinion, this study can be evaluated for publication after reconsidering the shortcomings mentioned above and the comments given below.

Thanks for your comments and good suggestions. Actually, long wavelength light (such as NIR region) photosensitizers is more suitable for PDT because of their deeper penetration. NIR photosensitizers without targeting ability could also be activated by sunlight because sunlight contains nonnegligible NIR part. However, the biotinylated photosensitizer mentioned in this manuscript possessed tumor selectivity and had little effect on normal cells (COS 7 cells) under white light, but could greatly inhibit the proliferation of tumor cells (MCF-7 cells), as shown in Figure 4. So, the patients could interact with sun light after treatment.

As put in the introduction, daylight photodynamic therapy (DL-PDT) has made PDT more widespread, cheaper, and less painful. Considering that we have synthesized three biotination of photosensitizers, their absorption wavelengths are not similar, so we use white light source uniformly. In fact, these photosensitizers can be activated by NIR light. For example, the absorption window of our photosensitizer such as DCF-TFM-Biotin is in the range of 600-700 nm (Figure 1a). Under the irradiation of 660 nm LED lamp (20 mW/cm^2), singlet oxygen (Figure R1a and b) and superoxide anion (Figure R1c and d) could still be produced.

Figure R1. UV-vis spectra of DPBF (50 μM) in the presence (a) and absence (b) of DCF-TFM-Biotin (10 μM) in ethanol under 660 nm light irradiation (20 mW cm⁻²). UV-vis spectra of NBT (25 μM) in the presence (c) and absence (d) of DCF-TFM-Biotin (10 μM) in PBS under 660 nm light irradiation (20 mW cm⁻²).

Some comments

1. What is the used excitation wavelength of the emission spectra given in Figure 1.

Thanks for your comments and good suggestions. The used excitation wavelength of the emission spectra given in Figure 1 is 640 nm, 469 nm and 403 nm, respectively. Relevant information has been added in the appropriate place in the revised manuscript.

2. a, b, c and d should be labelled in Figure 5.

Thanks for your comments and good suggestion. This error has been corrected in the revised manuscript.

3. Lines 220 and 221: “respectively” should be removed.

Thanks for your comments and good suggestion. This error has been corrected in the revised manuscript.

4. Line 246: in “10⁵” 5 should be superscript.

Thanks for your comments and good suggestion. This error has been corrected in the revised manuscript.

5. Line 256: “2% O₂” should be “1% O₂”.

Thanks for your comments and good suggestion. This error has been corrected in the revised manuscript.

6. What is the reaction temperature for the synthesis of DCF-TFM?

Thanks for your comments and good suggestion. The reaction temperature for the synthesis of DCF-TFM is 60 °C. Relevant information has been added in the appropriate place in the revised supporting information.

7. Totally 22 H were given in the ¹H NMR data of DCF-TFM but this compound contains 24 H.

Thanks for your comments and good suggestion. Totally 22 H were given in the ¹H NMR data of DCF-TFM, because hydrogen of phenolic hydroxyl group and carboxyl group is active, which is

easy to exchange with deuterium in deuterium solvent, so it does not appear.

8. I calculated the molecular weight of the compound DCF-TFM as 819.60. It should be checked.

Thanks for your comments and good suggestion. We checked the molecular weight of the compound DCF-TFM by ChemDraw. The exact mass is 818.1084. m/z: 818.1084 (100.0%), 820.1054 (63.9%), 819.1117 (47.6%), 821.1088 (30.4%), 820.1151 (11.1%), 822.1025 (10.2%), 822.1121 (7.1%), 823.1058 (4.9%), 819.1054 (2.2%), 820.1126 (1.4%), 821.1024 (1.4%), 824.1092 (1.1%), 820.1087 (1.1%).

9. Totally 13 H were given in the 1H NMR data of Biotin-NH2-NH2 but this compound contains 18 H.

Thanks for your comments and good suggestion. The compound Biotin-NH-NH₂ contains three amino hydrogens and two imide hydrogens. The five hydrogens are active, exchanged with deuterium in D₂O, so they don't show up. So totally 13 H were given in the 1H NMR data.

10. Totally 37 H were given in the 1H NMR data of DCF-TFM-Biotin but this compound contains 40 H.

Thanks for your comments and good suggestion. Hydrogen of phenolic hydroxyl group and two hydrogens of imide group don't appear in the 1H NMR data of DCF-TFM-Biotin.

11. I calculated the molecular weight of the compound DCF-TFM-Biotin as 1059.93. It should be checked.

Thanks for your comments and good suggestion. We checked the molecular weight of the compound DCF-TFM-Biotin by ChemDraw. The exact mass is 1058.2128. m/z: 1058.2128 (100.0%), 1060.2099 (63.9%), 1059.2162 (58.4%), 1061.2132 (37.3%), 1060.2195 (16.7%), 1062.2166 (10.7%), 1062.2069 (10.2%), 1063.2103 (6.0%), 1060.2086 (4.5%), 1059.2099 (3.7%), 1062.2057 (2.9%), 1061.2120 (2.6%), 1061.2069 (2.4%), 1061.2229 (2.3%), 1060.2132 (2.2%), 1063.2199 (2.0%), 1064.2136 (1.7%), 1063.2090 (1.7%), 1060.2171 (1.6%), 1062.2103 (1.4%), 1062.2141 (1.1%).

12. Totally 32 H were given in the 1H NMR data of FL-Biotin but this compound contains 38 H.

Thanks for your comments and good suggestion. ¹H NMR (400 MHz, DMSO-d₆) δ 10.52 (s, 1H), 9.96 (s, 1H), 8.10 (d, *J* = 7.9 Hz, 2H), 8.01 – 7.85 (m, 2H), 7.82 – 7.65 (m, 3H), 7.65 (s, 1H), 6.91 (s, 2H), 6.59 (s, 2H), 6.37 (s, 2H), 4.33 (t, *J* = 6.5 Hz, 1H), 4.18 (d, *J* = 5.1 Hz, 1H), 3.16 (d, *J* = 7.6 Hz, 1H), 2.84 (q, *J* = 6.8, 6.1 Hz, 2H), 2.31 (s, 6H), 2.09 – 1.87 (m, 2H), 1.73 – 1.51 (m, 4H), 0.89 – 0.76 (m, 2H). Totally 35 H were given in the 1H NMR data of FL-Biotin. Hydrogen of phenolic hydroxyl group and two hydrogens of imide group don't appear in the 1H NMR data of FL-Biotin.

13. I calculated the molecular weight of the compound FL-Biotin as 1005.88. It should be checked.

Thanks for your comments and good suggestion. We checked the molecular weight of the compound FL-Biotin by ChemDraw. The exact mass is 1004.1910. m/z: 1004.1910 (100.0%), 1006.1881 (63.9%), 1005.1944 (56.2%), 1007.1914 (36.0%), 1006.1977 (15.5%), 1008.1851 (10.2%), 1008.1948 (9.9%), 1009.1885 (5.7%), 1006.1868 (4.5%), 1005.1881 (3.0%), 1008.1839 (2.9%), 1007.1902 (2.5%), 1007.2011 (2.0%), 1007.1851 (1.9%), 1006.1914 (1.7%), 1006.1953 (1.6%), 1009.1872 (1.6%), 1010.1918 (1.6%), 1009.1982 (1.3%), 1008.1885 (1.1%), 1008.1923 (1.1%).

14. Figure S10 did not cite in the manuscript.

Thanks for your comments and good suggestion. Figure S10 corresponds to “MCTS was incubated with **DCF-TFM-Biotin** for 4h, the strong red fluorescence in the entire spheroid was observed, which indicates that the MCTS was completely infiltrated by **DCF-TFM-Biotin** (Figure S10).” The original manuscript mislabeled Figure S10 as Figure S11. This error has been corrected in the revised manuscript.

15. ¹³C NMR spectra of the compounds FL-Biotin and PpIX-Biotin should be supplied.

Thanks for your comments and good suggestion. ¹³C NMR spectra of the compounds FL-Biotin and PpIX-Biotin were added in the revised supporting information.

Reviewer 2

The authors reported three new photosensitizers for PDT studies. The biotination moiety was incorporated in these photosensitizers to enhance the tumor cell selectivity. As expected, the authors noticed that the newly designed sensitizers did show selectivity towards tumor cells over the normal cells. In addition, they found the biotination induced the generation of superoxide radical (O₂⁻), which eventually helps to increase the PDT effect in even hypoxia environments. This effect was attributed to the combination of type I and II mechanisms. Among the three photosensitizers, the DCF-TFM-Biotin was found to be ideal for practical applications. The paper is well written, and sufficient evidence was provided for every conclusion. Therefore, I believe this paper is suitable for publication in Nature Communications.

Minor comments:

1. How the superoxide radicals were produced when there is no oxygen around? I mean, under hypoxia conditions the oxygen levels are very low, so how can produce such large quantities of superoxide radicals?

Thanks for your comments and good suggestion. Many literatures have reported that Type-I PSs can lower oxygen-dependence by avoiding direct and fast O₂ depletion in PDT to solve the hypoxic problem.^{1, 2} In detail, formed superoxide radicals by the Type-I process participate in superoxide dismutase (SOD) -triggered catalytic cascades to produce O₂ for recycling. And type I PDT process can make full utilization of Haber–Weiss reaction, or Fenton reaction to compensate for O₂-depletion.^{3, 4} So, there are large quantities of O₂⁻ generation even in hypoxia conditions.

2. To me it looks like we need O₂ for both types of I and II mechanisms. Is there another pathway for the production of radical (SOD) in Type I mechanism?

Thanks for your comments and good suggestion. The reviewer is right that both types of I and II mechanisms do need oxygen. Type I PDT, which activates free radicals from various substrate molecules, but not limited to O₂. Superoxide anion formed by Type-I process can be used as the initiator of other reactive oxygen species. O₂⁻ could be catalyzed by intracellular SOD and transformed into other highly cytotoxic radicals (e.g., OH•) through Haber-Weiss reaction or Fenton reaction as mentioned in Reply 1.

3. How the authors avoided double biotination of PpIX?

Thanks for your comments and good suggestion. We strictly controlled the feeding of PpIX with EDCI, HOBt, and DIEA according to the molar ratio of 1:1:1:1, so that the carboxyl group on one side of PpIX could be esterified as much as possible, and then Biotin-NH-NH₂ was added drop by drop (PpIX: Biotin-NH-NH₂ = 1:0.86). We clarify it in the revised manuscript.

4. Must show the entire NMR range (Figure S29), from -4 ppm to 10 ppm. Must include inner NH peaks.

Thanks for your comments and good suggestion. the entire NMR range (Figure S29) is shown in Figure R2. inner NH peaks are δ 10.15 and δ 6.45-6.39.

Figure R2. ¹H-NMR spectrum of compound PpIX -Biotin in (CD₃)₂SO.

Reviewer 3

The current paper is part of an ongoing research carried out by the authors on the development of organic photosensitizers, and their application in photodynamic therapy (PDT). More specifically, this article is directed toward the development of organic photosensitizers for PDT type I, in order to overcome hypoxia considered as the Achilles heel of PDT. The authors' basic finding, is that the biotination of 3 organic photosensitizers increases their ability to generate superoxide anion O₂⁻. making them interesting candidates for type 1 PDT. Although this finding is quite interesting and in itself supports the publication of the article, there are some points that authors should address before accepting this paper in this journal or elsewhere:

- Although the reported effect of biotination was unexpected, it nevertheless has precedents. Surprisingly, one of the most representative precedent was reported by some of the authors. Indeed, in a full paper published recently in JACS, Peng's group reported that the biotination of ENBS afforded ENBS-B a photosensitizer with a better capacity for O₂⁻ generation. This article (reference 30), inadequately quoted by the authors, should be cited in the introductory part and the structure of ENBS-B should also be given.

Thanks for your comments and good suggestion. We reintroduce the work of Peng et al in the introduction of revised manuscript, especially mention the name of the photosensitizer ENBS-B.

- Apart from Peng's article, the biotination of other photosensitizers has already been reported in

the literature, and should be cited (New J. Chem. 2020, 44, 3392-3401; J.Mater.Chem.B, 2019,7,65-79; Chem. Commun., 2019, 55, 10972-10975; J.Photochem. Photobiol.,B, 2019, 190, 1-7)

Thanks for your comments and good suggestion. The mentioned literatures have been cited in the revised manuscript.

- On the other hand, the authors claim that PDT type I based on organic photosensitizers is very rare. Being true some few years ago, the panorama is drastically changing as shown by the increasing number of works reports on this subject. Most of these advances are included in the recently released review by Chen et al. "Type I Photosensitizers Revitalizing Photodynamic Oncotherapy. Small 2021, 17, 2006742".

Thanks for your comments and good suggestion. Type I photosensitizers based on organic molecules have been developing rapidly in recent years. The literature (Small 2021, 17, 2006742) was cited in the revised manuscript. And we revised the description in the introduction.

- In the cell selectivity studies, the authors used MCF7 breast cancer cells and kidney Cos-7 cells. Would it not be more accurate to use in this study non-malignant breast cells MCF10 usually used as a reliable model for normal mammary cells?

Thanks for your comments and good suggestion. If possible, using MCF10 would be better for comparison. However, it was found that MCF10 cells were less used than COS7 cells, because the former is much more difficult to cultivate than the latter. COS 7 was widely used in the literatures as a comparison object for MCF7 or other tumor cells and did not affect the experimental results.⁵⁻⁹

- The authors must compare the efficiency of their system with the commercially available, clinically-used photosensitizer chlorin e6

Thanks for your comments and good suggestion. In 2019, our group has studied the phototoxicity of protoporphyrin (PpIX, another clinically-used photosensitizer, similar to chlorin e6) under different oxygen-containing conditions.¹⁰ The inhibition rate of PpIX on tumor cells under 1% oxygen condition was only 20% at high concentration (20 μ M) (ACS Appl. Mater. Interfaces 2019, 11, 15426-15435, Figure 4a), far lower than the 60% inhibition rate of DCF-TFM-Biotin (manuscript Figure 4b). The photocytotoxicity of **DCF-TFM-Biotin** was mainly attributed to the generated $O_2^{\cdot-}$ through Type I pathway.

- The authors must carry out in vivo studies in order to show the effectiveness of their system in terms of activity, bioavailability and targeting.

Thanks for your comments and good suggestion. Relevant experiments have been supplemented in the revised manuscript and supporting information.

- The authors must also conduct mechanistic studies to rationalize the observed effect.

Thanks for your comments and good suggestion. A mechanistic explanation of the biotination effect is added in the revised manuscript. Type I process is that the triplet PS is transformed into a radical anion by accepting electrons from adjacent substrates and giving external electrons to oxygen to form $O_2^{\cdot-}$. Photosensitizer has a lower reduction potential demonstrating the stronger electron-accepting character.^{1, 11, 12} We studied the electrochemical properties of **DCF-TFM**, **DCF-TFM-Biotin**, **FL** and **FL-Biotin** by cyclic voltammetry with Ferrocene (Fc) as the external standard. As shown in Figure 10, the reduction event of **DCF-TFM-Biotin** at -0.879 V is lower

than -1.072V of **DCF-TFM**. Similarly, the anodic shift of biotination of photosensitizers facilitate them to accept electrons, which endow **DCF-TFM-Biotin**, **FL-Biotin** and **PpIX-Biotin** with the potential to produce more $O_2^{\cdot-}$ by the Type-I process.

Reference

1. Teng, K.X. *et al.* BODIPY-Based Photodynamic Agents for Exclusively Generating Superoxide Radical over Singlet Oxygen. *Angew Chem Int Ed Engl* **60**, 19912-19920 (2021).
2. Bu, Y. *et al.* A NIR-I light-responsive superoxide radical generator with cancer cell membrane targeting ability for enhanced imaging-guided photodynamic therapy. *Chem Sci* **11**, 10279-10286 (2020).
3. Novohradsky, V. *et al.* Towards Novel Photodynamic Anticancer Agents Generating Superoxide Anion Radicals: A Cyclometalated Ir(III) Complex Conjugated to a Far-Red Emitting Coumarin. *Angew Chem Int Ed Engl* **58**, 6311-6315 (2019).
4. Li, M. *et al.* Near-Infrared Light-Initiated Molecular Superoxide Radical Generator: Rejuvenating Photodynamic Therapy against Hypoxic Tumors. *J Am Chem Soc* **140**, 14851-14859 (2018).
5. Daniel Pedro-Hernandez, L., Hernandez-Montalban, C., Martinez-Klimova, E., Ramirez-Apan, T. & Martinez-Garcia, M. Synthesis and anticancer activity of open-resorcinarene conjugates. *Bioorg Med Chem Lett* **30**, 127275 (2020).
6. Sánchez-Mora, A. *et al.* NHC-Ir(I) complexes derived from 5,6-dinitrobenzimidazole. Synthesis, characterization and preliminary evaluation of their in vitro anticancer activity. *Inorganica Chimica Acta* **496** (2019).
7. Gebremedhin, K.H. *et al.* Benzo[a]phenoselenazine-based NIR photosensitizer for tumor-targeting photodynamic therapy via lysosomal-disruption pathway. *Dyes and Pigments* **170** (2019).
8. Gurram, B. *et al.* Celecoxib Conjugated Fluorescent Probe for Identification and Discrimination of Cyclooxygenase-2 Enzyme in Cancer Cells. *Anal Chem* **90**, 5187-5193 (2018).
9. Wang, X.Q. *et al.* Cucurbit[8]uril Regulated Activatable Supramolecular Photosensitizer for Targeted Cancer Imaging and Photodynamic Therapy. *ACS Appl Mater Interfaces* **8**, 22892-22899 (2016).
10. Liu, Z. *et al.* Nitroreductase-Activatable Theranostic Molecules with High PDT Efficiency under Mild Hypoxia Based on a TADF Fluorescein Derivative. *ACS Appl Mater Interfaces* **11**, 15426-15435 (2019).
11. Zhuang, Z. *et al.* Type I photosensitizers based on phosphindole oxide for photodynamic therapy: apoptosis and autophagy induced by endoplasmic reticulum stress. *Chemical Science* **11**, 3405-3417 (2020).
12. Huang, H. *et al.* Targeted photoredox catalysis in cancer cells. *Nat Chem* **11**, 1041-1048 (2019).

REVIEWER COMMENTS

Reviewer #1 (Remarks to the Author):

I checked the responses to the critics given by author again (especially given to Reviewer #3). I think the authors give acceptable responses to the critics given by Reviewer #3. They cited all of references suggesting by Reviewer #3. As far as I look the authors only did not compare the efficiency of their system with the commercially available, clinically-used photosensitizer chlorin e6.

On the other hand, the sentence "As far as we know, there is few purely organic photosensitizers reported to perform Type I & Type II PDT under white light irradiation" given in the introduction part should be removed because reported purely organic photosensitizers are not rare.

As a result, I think the authors were answered adequately to the Reviewers critics.

Reviewer #2 (Remarks to the Author):

My initial comment # 4 was not addressed adequately. Authors should review porphyrin literature to familiarize the NMR spectra of porphyrins. Authors mentioned that inner NH peaks appear at δ 10.15 and δ 6.45-6.39. This is completely wrong! They should appear at \sim 3.00 ppm.

Reviewer #1 (Remarks to the Author):

I checked the responses to the critics given by author again (especially given to Reviewer #3). I think the authors give acceptable responses to the critics given by Reviewer #3. They cited all of references suggesting by Reviewer #3. As far as I look the authors only did not compare the efficiency of their system with the commercially available, clinically-used photosensitizer chlorin e6.

Thanks for your comments and good suggestions. In the manuscript we provided three biotinylated photosensitizers, of which PpIX-Biotin was derived from the commercial photosensitizer PpIX. PpIX as a typical type II photosensitizer is very similar to chlorin e6 in terms of the structure (shown in Fig. R1) and optical properties^{1,2}. It is a good choice to compare PpIX-Biotin with PpIX. In addition, we have evaluated the photocytotoxicity and dark cytotoxicity of PpIX and PpIX-Biotin under normoxia and hypoxia, respectively, in the revised manuscript (Figure S10). It was found that PpIX-Biotin possessed a better photodynamic effect both under normoxia and hypoxia than PpIX because it could produce both $^1\text{O}_2$ and O_2^- , and the type I mechanism could still function well under depleted oxygen. Meanwhile, the photodynamic effect of DCF-TFM-Biotin under hypoxia was indeed superior to that of PpIX. And PpIX-Biotin exhibited the best PDT effect in both hypoxia and normoxia, which is the reason that PpIX-Biotin was chosen to demonstrate the in vivo PDT effect. .

Fig.R1 The chemical structures of PpIX and chlorin e6.

On the other hand, the sentence “As far as we know, there is few purely organic photosensitizers reported to perform Type I & Type II PDT under white light irradiation” given in the introduction part should be removed because reported purely organic photosensitizers are not rare.

Thanks for your comments and good suggestions. We removed the sentence “As far as we know, there is few purely organic photosensitizers reported to perform Type I & Type II PDT under white light irradiation” in the revised manuscript.

Reviewer #2 (Remarks to the Author):

My initial comment # 4 was not addressed adequately. Authors should review porphyrin literature to familiarize the NMR spectra of porphyrins. Authors mentioned that inner NH peaks appear at δ 10.15 and δ 6.45-6.39. This is completely wrong! They should appear at \sim -3.00 ppm.

Thanks for your comments and good suggestions. You are right, the attribution of inner NH peaks was incorrect. When The signals in the ^1H NMR of PpIX-Biotin were reassigned in the revised manuscript (as shown in Figure S28). Inner NH peaks appear at δ -4.17, which is similar to the inner NH of PpIX (δ -4.24, shown in Fig.R2). And totally 47 H were given in the ^1H NMR, because -COOH (1H), and -CONH (2H) didn't appear. Last time, the NMR spectra was recorded

from -4 ppm to 10 ppm. And we missed the peaks of inner NH which is beyond -4 ppm. We guess that the peak is at around -4.2 ppm not -3.0 ppm is because the used NMR solvents are different.

Fig.R2 The ^1H NMR spectrum of PpIX in $(\text{CD}_3)_2\text{SO}$.

1. Rinco, O. *et al.* The effect of porphyrin structure on binding to human serum albumin by fluorescence spectroscopy. *Journal of Photochemistry and Photobiology A: Chemistry* **208**, 91-96 (2009).
2. Ormond, A.B. & Freeman, H.S. Dye Sensitizers for Photodynamic Therapy. *Materials (Basel)* **6**, 817-840 (2013).